
# Saint Petersburg magnetic observatory: from Voeikovo subdivision to INTERMAGNET certification

Roman Sidorov[1], Anatoly Soloviev[1,2], Roman Krasnoperov[1], Dmitry Kudin[1,3], Andrei Grudnev[1,2], Yury Kopytenko[4], Andrei Kotikov[4,5], Pavel Sergushin[4]

[1]Geophysical Center of the Russian Academy of Sciences (GC RAS), 119296 Moscow, Russian Federation
[2]Schmidt Institute of Physics of the Earth of the Russian Academy of Sciences (IPE RAS), 123242 Moscow, Russian Federation
[3]Laboratory of Robot Technique, Gorno-Altaisk State University (GASU), 649000 Gorno-Altaisk, Altai Republic, Russian Federation
[4]Saint Petersburg branch, Pushkov Institute of Terrestrial Magnetism, Ionosphere and Radio Wave Propagation of the Russian Academy of Sciences (IZMIRAN), 199034 Saint Petersburg, Russian Federation
[5]Department of Physics of the Earth, Saint Petersburg State University, 199034 Saint Petersburg, Russian Federation

*Correspondence to*: Roman Sidorov (r.sidorov@gcras.ru)

**Abstract.** Since June 2012 the Saint Petersburg magnetic observatory is being developed and maintained by two institutions of the Russian Academy of Sciences (RAS) – the Geophysical Center of RAS (GC RAS) and the Saint Petersburg branch of the Pushkov Institute of Terrestrial Magnetism, Ionosphere and Radio Wave Propagation of RAS (IZMIRAN SPb Branch). On 29 April 2016 the application from the Saint Petersburg observatory (IAGA code SPG) for introduction into the INTERMAGNET network was accepted after approval by the experts of the first definitive data set over 2015, produced by the GC RAS, and on 9 June 2016 the SPG observatory was officially certified. One of the oldest series of magnetic observations originated in 1834 was resumed in the 21st century, meeting the highest quality standards and all modern technical requirements. In this paper a brief historical and scientific background of the SPG observatory foundation and development is given, the stages of its renovation and upgrade in the 21st century are described, and information on its current state is provided. The first results of the observatory functioning are discussed, geomagnetic variations registered at the SPG observatory are assessed and compared with geomagnetic data from the INTERMAGNET observatories, located in the same region.

## 1 Historical background

First observations of the Earth's magnetic field elements and their variations in the vicinity of Saint Petersburg date back to 1726. In 1834 a regularly functioning Russian network for geophysical observations was established (Pasetsky and Svetlaev, 1978). After the Saint Petersburg Main Physical Observatory became the part of the Academy of Sciences, the prominent Russian physicist, chemist and metrologist Adolph-Theodor Kupffer introduced a project of establishing a magnetic and meteorological observatory located outside the city in 1865. This plan was implemented in 1876–1878, and in 1878 the



Pavlovsk observatory started functioning on a regular basis. A continuous series of magnetic measurements at the Pavlovsk observatory was maintained from 1878 to 1941. By the end of World War II the magnetic measurements were continued at the magnetic and meteorological observatory Voeikovo, operated by the Leningrad branch of the Research Institute of Terrestrial Magnetism (currently – IZMIRAN).

In the late 1960s a magnetic station Krasnoe Ozero (literally, the "Red Lake") was established in the Vyborg district of the Leningrad Region, 100 km northwest from the city of Leningrad (currently – Saint Petersburg) and 89 km southeast from the city of Vyborg (Fig. 1). This station initially was the branch of the Voeikovo observatory. It was deployed for relocation of magnetometric equipment and instruments from Voeikovo since the level of anthropogenic disturbances had become unacceptable for proper observations. The Krasnoe Ozero station operated till 2000. In 2010, GC RAS and IZMIRAN SPb

Branch agreed to deploy a new collaborative high-quality magnetic observatory on the basis of the Krasnoe Ozero magnetic station in the course of development of the Russian segment of the INTERMAGNET network (Soloviev et al., 2013; Gvishiani et al., 2014; Gvishiani and Lukianova, 2015). With the joint efforts of these two institutions of the Russian Academy of Sciences this project has become a new milestone in the history of magnetic measurements in this region. This new INTERMAGNET-standard observatory was designated the official name "Saint Petersburg" and IAGA code SPG.

## 2 Observatory deployment

In 2012 the process of renovation of the Krasnoe Ozero magnetic station and deployment of the SPG observatory were initiated.

### 2.1 Magnetic survey of the territory

The SPG deployment was preceded by a detailed magnetic survey of the station's territory for assessment of its magnetic
characteristics and detection of possible sources of magnetic disturbances. The survey consisted of measuring the magnetic anomalies and magnetic field vertical gradient. The vertical gradient of the total magnetic field vector was determined as the ratio of difference in readings between two magnetic gradiometer sensors mounted vertically above each other to the distance between the sensors' axes (in most cases it was 56 cm – the length of one standard rod section between the sensors). Magnetic gradiometry allows to reveal spatially small disturbances against the background anomalies related to geological
inhomogeneities. A gradiometer (GEM Systems GSM-19GW was used) also provides elimination of time variations of the field during the survey, estimating only spatial effects. In addition, the usual modification of a GEM Systems GSM-19 scalar magnetometer was installed as a base station for compensation of the magnetic field diurnal variations that could affect the survey interpretation.

The magnetic survey points within a 4100 $m^2$ area surrounding the pavilions were set out for the magnetic survey using an
30 optical theodolite and a 50 m geodetic measuring tape. The overall length of survey lines was 480 m (56 magnetic survey points) with a spatial resolution of 10×10 m, as the interval between the survey points and the distance between the survey





lines was 10 m. This was considered as an optimal spatial resolution to reveal possible heterogeneities the distribution of anomalies of the total magnetic field and its vertical gradient. The overall error of setting out the survey points was ~20 cm, i.e. 2% of the spatial resolution of the survey which was considered negligible. The survey lines were marked along a North-South direction. The built-in GNSS-receiver of the gradiometer was used to determine the survey point coordinates. After

the survey, the measurements were processed on a PC, where the recorded data were imported from the magnetic gradiometer and the base magnetometer. Time variations (diurnal variation, pulsations, etc.), which occurred during the survey, were compensated while processing. For this purpose, the gradiometer data recordings were corresponded in time with the ones recorded by the base magnetometer, so that the time of the registration for every recording of the magnetic gradiometer coincided or was the nearest to the time of the registration for the base magnetometer data. Thus, the anomalous

component $\Delta F_a$ of the total magnetic field intensity at each observation point was calculated using the following Eq. (1):

$$\Delta F_a(i) = F_s\big(t_s(i)\big) - F_b\big(t_b(j)\big) \tag{1}$$

where $F_s$ is the survey magnetic recording, $F_b$ is the base magnetometer recording, and the time moment of a survey data recording $t_s(i)$ and the one of a base magnetometer recording $t_b(j)$ produce the minimum of the difference $|t_s(i) - t_b(j)|$ (including zero if they match second to second).

After that the maps of the anomalous component and the vertical gradient of total field were plotted. We used triangulation with linear interpolation for gridding the survey data along a regular grid.

The map analysis showed that the territory, surrounding the pavilions, was generally characterized by homogeneous field. A strong magnetic anomaly on the west of the $\Delta F_a$ plot (Fig. 2a) results from certain gardening equipment which didn't affect the overall magnetic field distribution significantly (later the source of anomaly was removed to provide even more magnetic

cleanliness). The lateral spatial variability of the magnetic field was considered insignificant in the vicinity of the pavilions. The distribution of the vertical gradient values over the survey plot (Fig. 2b) is represented mainly by zero values. Therefore, the results of the survey showed that the area of the Krasnoe Ozero station was suitable for the installation of the INTERMAGNET-standard equipment for registering the total magnetic field, its variations and carrying out absolute measurements. During this stage preliminary absolute measurements were performed as well to estimate the characteristic

values of magnetic declination and inclination within the station's site.

## 2.2 Observatory infrastructure renovation

By the end of 2012 the interior of the station's pavilions was renovated (Fig. 3), a new heating system was installed, and the instrument pillars were fixed. Initially the Krasnoe Ozero station had a water heating system based on water supply via copper pipelines. In the absolute pavilion, a demountable wooden screen with an illuminator was placed over the window for

directed sighting the azimuth mark. This helps to regulate the thermal balance and avoid refraction due to the temperature contrast when opening the window in cold seasons.



## 2.3 Installation of magnetometric instruments

In 2012 a full set of magnetometric instruments of the INTERMAGNET standard was installed into the observatory pavilions. It includes a DTU Space FGE vector fluxgate magnetometer, a GEM Systems GSM-19 proton scalar Overhauser magnetometer, a Mingeo fluxgate declinometer/inclinometer, based on Carl Zeiss Theo010 non-magnetic theodolite, and a

Mingeo Magrec data acquisition system.

The DTU Space FGE vector fluxgate magnetometer was installed on a pillar at the variation pavilion. This magnetometer is equipped with a 24-bit AD converter and provides digital registration of measurements at the frequency up to 10 Hz. Positioning and adjusting of the magnetometer's sensor system with reference to the geographic coordinate system was considered more preferable. Although the process of the sensors' alignment in the magnetic coordinate system is often

recommended by various researchers as a relatively easy alternative, it requires future recurrent adjustments of the sensor direction due to variability of the magnetic North pole coordinates. Thus, the measured values for the vector magnetometer correspond to the variations of the magnetic field components in the northern (X), eastern (Y), and vertical (Z) geographical directions. The absolute values of the magnetic field vector components were calculated based on the performed absolute observations.

The GEM Systems GSM-19 scalar magnetometer sensor was mounted on the top of a pillar inside the absolute pavilion in a wooden cradle and fixed. The GSM-19 sensor axis was oriented horizontally and normal to the magnetic meridian plane. The fluxgate declinometer/inclinometer was mounted on another pillar in the absolute pavilion.

The observatory data acquisition system, installed in the main building, was configured for transmission of data in the near-realtime mode to GC RAS, IZMIRAN and later to the INTERMAGNET GIN in Paris. Low voltage (12 V) batteries supply

power to the scalar, vector magnetometers and the Magrec data acquisition system via underground power lines.

In 2015 a series of improvements in data transmission were implemented by GC RAS specialists. A new antenna for amplification the 3G internet connection was mounted, and a new 3G-modem was installed directly into the Magrec data logger. The latter was set up to provide remote access and control. Also in autumn 2015 certain improvements were made to achieve a better data quality. First, the sources of distortion of the magnetic records from the FGE vector magnetometer were

eliminated. Next, the total check of electric connections, hardware cables and the heating system was done at the variation pavilion. Finally, the software for the data acquisition system computer was updated.

## 2.4 Azimuth mark installation and reference azimuth determination

In 2012 a new observatory azimuth mark (or mira) was constructed. The requirements for installing the reference azimuth mark for absolute measurements are significantly important since the measurements of magnetic declination and inclination

require direct visibility of the azimuth mark or a remote benchmark. As it is commonly not possible to use a fundamental structure in 1–2 km from the absolute pavilion as an azimuth mark, one should consider that the less the distance between the





mark and the observation point, the firmer it should be fixed. For example, the shift of the azimuth mark installed in 100 m from the observation point shouldn't be more than 3 mm laterally.

The azimuth mark for the SPG observatory was developed, assembled and tested at the Voeikovo observatory and installed at SPG in autumn 2012. It is a steel construction equipped with a light bulb for carrying out the absolute measurements in

case of possible poor visibility and lack of light due to weather conditions. The mark is mounted on four supports onto a concrete basement providing high stability (Fig. 4a). The mark can be clearly seen from the measurement pillar through the theodolite telescope (Fig. 4b).

In the manuals dedicated to ground magnetic observatory practice (Jankowski and Sucksdorff, 1996; Nechaev, 2006), it is recommended to determine the reference azimuth for absolute measurements by carrying out astronomical observations.

Azimuth determination from Sun observations ensures an astronomical azimuth value with an error of about dozens of arc seconds. However, such approach is significantly labor consuming and also requires high-accuracy geodetic tools and a qualified operator. With the development of the global navigation satellite systems' (GNSS) technologies, it became possible to determine the reference direction at magnetic observatories without the mentioned disadvantages. A method of determining coordinates was recommended in a guide (Newitt et al., 1996). It provides the positioning of a station within a

2 cm accuracy and determining of geodetic azimuths of reference directions on distances of several hundred meters with an accuracy of several arc seconds. The accuracy of azimuth determination depends on the length of the baseline: the longer the baseline direction, the higher the accuracy of the reference azimuth measurement. This method has already been implemented at magnetic repeat stations abroad (Lalanne et al., 2013) but it has not yet become a wide practice at geomagnetic observatories in Russia. The approach for azimuth determination of reference directions at magnetic

observatories, based on modern geodetic equipment and technologies, has been successfully introduced and implemented by the GC RAS' specialists (Kaftan and Krasnoperov, 2015; Krasnoperov et al., 2015).

The geodetic equipment that was used for the measurements included two sets of GPS Javad Maxor receivers with Legant antennae and a Trimble M3 DR5" electronic laser total station with a standard prism reflector. The GPS receivers were positioned at auxiliary points for determining the azimuth of the baseline between these points. The total station and the

25 prism reflector were used to transmit the geodetic azimuth to the reference direction of the mark. In the conditions of forestation and other obstacles for mutual visibility between the points on the territory of the SPG observatory, it was impossible to obtain a baseline longer than 150 m; however, the accuracy of determining the azimuth of the reference direction was 2–3 seconds of arc (Kaftan and Krasnoperov, 2015), which meets the INTERMAGNET requirements for the azimuth mark given in (Benoit, 2012; Jankowski and Sucksdorff, 1996; Newitt et al., 1996). Also, the coordinates of the

30 pillar centers at the absolute pavilion of the observatory were obtained for the first time with reference to the common international coordinate reference frame.



## 3 Observatory data analysis and discussion

### 3.1 Variation data analysis

To make sure that the correct functioning of the magnetometers and proper variation data quality is provided, we made a qualitative and quantitative comparison between the data records registered at the SPG observatory and the ones from the

Borok (BOX, Russia), Lerwick (LER, UK) and Nurmijarvi (NUR, Finland) observatories. These three INTERMAGNET observatories are the closest to the SPG observatory both by geographic and magnetic latitudes (see Table 1 for details on their geographic locations etc.). We selected three time periods corresponding to different space weather and solar activity conditions to estimate the geomagnetic variation signal forms during quiet and disturbed periods, as it can provide a more representative overview of the magnetometers' operation comparing to the ones at other observatories.

The periods were chosen for this research according to the Kp index data (Kp-Index, 2017). We chose two disturbed periods in March and June 2015 to estimate the response of the magnetometers to two geomagnetic storms occurred during the period that later was selected for the INTERMAGNET certification of the SPG observatory. On 17–18 March 2015 the strongest geomagnetic storm of the current Solar cycle took place. During this storm the planetary Kp index values varied from 4+ to 8–, and the Kp sum was 48o for the $1^{st}$ day of the storm and 39+ for the second day (here we use common

notations for Kp magnitudes). Its analysis based on SPG data is discussed in (Gvishiani et al., 2016). Another storm occurred on 22–24 June 2015, and it was also a large geomagnetic activity disturbance: the Kp values varied from 3+ to 8+, and the Kp sum was 35+ for 22 June and 42 – for 23 June. We also compared the geomagnetic records for a quiet geomagnetic activity period and selected the INTERMAGNET data available for such period in June 2016. We considered a period 1–4 June 2016 represented by Kp index values within the range of 0o to 2o.

The horizontal components of the total field vector are commonly the most affected by magnetic disturbances, so in this paper we demonstrate the comparison plots only for the X component as the most illustrative. The data plots for the mentioned periods are given in Fig. 5–7. It is clearly seen that the SPG vector magnetometer records for the disturbed periods generally match the ones for these periods from the other observatories. Maximal signal similarity for X records can be seen between SPG (Fig. 5a and Fig. 6a) and BOX (Fig. 5b and Fig. 6b), as well as between SPG and NUR (Fig. 5d and 25

Fig. 6d). The similarity between BOX and LER (Fig. 5c and Fig. 6c) is not so obvious. Probably this is due to the location of LER observatory and its geomagnetic latitude (61.67°N for 2015 and 61.65°N for 2016) which is higher than the geomagnetic latitude for other two observatories from our list (about 53–57°N), although geographical latitude (60.13°N) is very close to the one for SPG (60.542°N). The difference in geomagnetic latitude means the difference in geomagnetic conditions due to a higher impact of auroral oval dynamics in the region of the observatory location. However, it is still

possible to visually match the signal fragments between SPG and LER X components corresponding to storm sudden commencements, onsets, main phases and the start moments of recovery phase. The data plots for quiet geomagnetic conditions (1–4 June 2016) again demonstrate high signal similarities between SPG (Fig. 7a) and BOX (Fig. 7b) and between SPG and NUR (Fig. 7d) variation data and lower similarity between SPG and LER (Fig. 7c) data.





For a quantitative assessment of the data similarity we estimated the Pearson linear correlation coefficient between SPG X, Y, Z and F records and the corresponding records from other three observatories during the abovementioned time periods. The correlation coefficient values are listed in Table 2. The correlation values prove the visual similarities of the data plots. Maximum correlation coefficient values are between SPG and NUR observatory (0.96–0.99). No scalar data for NUR observatory was available during all the analyzed periods. Also high correlation values were obtained for SPG and BOX observatories. The comparison for SPG and LER observatories resulted in the lowest correlation coefficient values, though still indicating a strong positive correlation (0.65–0.72). Hence, the quantitative comparison again shows the highest similarity between SPG and NUR observatory data due to their geographical and geomagnetic latitudinal closeness. Data records for another geomagnetically close observatory – Borok (BOX) – also have similarities, and the variation data from LER observatory has some differences comparing to SPG data because of differences in geomagnetic location and, therefore, differences in geomagnetic conditions.

Thus, the results of the comparison between the variation data from the SPG observatory and the corresponding variation datasets from three abovementioned geographically close INTERMAGNET observatories generally confirm that the magnetic equipment of the observatory is correctly installed, the vector magnetometer is properly aligned in the geographical coordinate system, and that there are no significant sources of regular electromagnetic noise in the vicinity of the observatory site.

Next, let us discuss the first definitive dataset for the SPG observatory as the main and final proof of its quality and suitability for the INTERMAGNET network in all aspects concerning the requirements formulated for its magnetic observatories (Benoit, 2012).

## 3.2 Absolute, baseline and definitive data analysis

The most important problem of magnetic observatories includes registering the Earth's magnetic field secular variation over long time periods (decades, centuries and even longer periods). To date, vector measurements of magnetic variations using fluxgate magnetometers similar to DTU Space FGE (Denmark) and IPGP VM391 (France) are carried out at many observatories. These vector variation measurements possess an unavoidable and unpredictable (Chulliat and Anisimov, 2008) drift on such long intervals mainly because of temperature variations, pillar and sensor deterioration, as well as instrument aging. Therefore, there is a need to perform regular measurements of the absolute magnetic declination, inclination and total intensity with a fluxgate magnetometer mounted on a non-magnetic optical theodolite (DIflux magnetometer) and a scalar magnetometer for vector data calibration (Jankowski and Sucksdorff, 1996).

Regular absolute measurements started at the SPG observatory in 2013 and they are taken at least once a week. The adopted baseline values are calculated for the X, Y, and Z components by the algorithm of fitting the spline curve to the observed baseline values resulted from the absolute measurements. Both, the observed and adopted baselines in 2015 are presented in Fig. 8. Two baseline splits in January and February 2015 were due to some improvement and repair works at the observatory. The quality of absolute measurements was estimated by calculating the root-mean-square (RMS) deviation of



differences between observed and adopted baseline values for each component. Evident absolute observation outliers were eliminated when spline approximation was applied. 2015 definitive data set was prepared at the Russian-Ukrainian Geomagnetic Data Center (2017) with the use of special software toolkit designed as a part of the automated hardware and software multi-functional system for geomagnetic monitoring (Gvishiani et al., 2016). The reliability of adopted baseline

application can be also estimated by comparing the obtained absolute measurements of the field components and baseline-corrected 1-minute values taken for the corresponding time moments. The obtained RMS deviations for baselines for the period 01.01.2015–01.01.2016 were 2.91, 2.08 and 0.61 nT for X, Y, and Z, respectively. The ranges of adopted baseline value variability were from –7.98 to 18.24 nT, from –7.09 to 15.86 nT, and from –0.78 to 4.23 nT for X, Y, and Z component, respectively. The corresponding dispersion plots are given in Fig. 9a, b, c.

Also the $\Delta F$ differences were analyzed to confirm 2015 definitive data quality. The $\Delta F$ values are the differences between the total field vector magnitudes directly measured with a scalar magnetometer and calculated from the vector measurements after their correction, using the adopted baselines. The complete $\Delta F$ series over 2015 is given in Fig. 9d. It is seen that the $\Delta F$ values variability is within 5 nT during a year, which meets the INTERMAGNET requirements (Benoit, 2012).

Finally, we compared definitive values of magnetic field components averaged over several magnetically quiet periods (1-2
January and 3-7 February 2015) with those derived from the internal magnetic field models. We used freely available and widely used models, which are considered to be the most precise at the moment: CHAOS-6 (Finlay et al., 2016), SIFM (Olsen et al., 2015) and EMM2015 (https://ngdc.noaa.gov/geomag/EMM/). All of them are constructed using the high-precision new generation satellite observations. Some of them include both the contributions of main and lithospheric field (see Table 3). CHAOS-6 and SIFM were used to calculate components for the 2015.1 epoch, and EMM2015 was used to
calculate components for the 2015.0 epoch, as it is the upper time limit for the latter model. All the model values were calculated for the SPG observatory geodetic location; local geomagnetic coordinates were reduced to local nominally geodetic coordinate system of magnetic measurements at the observatory. The comparison results generally indicate that the observed and modeled component values are in good agreement.

## 4 Conclusions

The SPG observatory was founded as a subdivision of the Voeikovo observatory, and in the beginning of the 21[st] century it became an autonomous scientific facility providing continuous geomagnetic measurements fully meeting the highest international quality standards. Qualitative and quantitative comparison between the SPG observatory variation and total field data with the data from three other INTERMAGNET observatories (BOX, LER and NUR), which are located close to it, was done. The comparison demonstrated the signal form similarity and relatively high correlation coefficients between the
SPG and the closest NUR observatory data both for quiet and disturbed periods. This indicates that the vector magnetometer at the SPG observatory is properly installed and aligned and the data are not affected by regular anthropogenic disturbances. Analysis of definitive data calculated from the SPG observatory variation and absolute data over 2015 using automated



software, integrated into the GC RAS's hardware and software system for geomagnetic monitoring (Gvishiani et al., 2016), showed that the resulting data records meet the INTERMAGNET standards.

Previously, the Russian INTERMAGNET segment included nine magnetic observatories: Arti (ARS), Borok (BOX), Irkutsk (IRT), Khabarovsk (KHB), Magadan (MGD), Novosibirsk (NVS), Paratunka (PET), Yakutsk (YAK), and Vostok (VOS).

The SPG observatory, recently accepted into the INTERMAGNET network, now also provides 1-second magnetic data with a near real-time transmission. The inclusion of this new observatory into the INTERMAGNET network contributes to accurate modelling of rapid and long-term variations of the core magnetic field, analysis of geomagnetic disturbances caused by the external magnetic field, their dynamics and spatiotemporal features. A particular advantage of the SPG observatory is its high-latitude location. Auroral and subauroral regions are mostly subjected to space weather, and therefore magnetic

observatories located in these regions are the most valuable source of information for monitoring and forecasting the space weather effects at the Earth's surface. High-latitude INTERMAGNET-standard observatories provide physically reliable datasets for detailed analysis of interplanetary environment and, thus, can be used for calibrating other types of ground-based observation systems, such as neutron monitors and muon hodoscopes. In particular, recent studies demonstrate that observatory records along with neutron monitor data and muonic hodoscope images can be used in the development of

ground-based distant techniques for early warning of geomagnetic storms (Hafez and Ghamry, 2013; Wu, 1997).

Digital object identifiers were assigned to the whole observatory database (Soloviev 2016), to its preliminary (Soloviev et al., 2016a) and 2015 definitive (Soloviev et al., 2016b) datasets. These data are considered as fully valid results of research and can be cited on the same basis as other scientific references.

## 5 Author contribution

The collaboration of the GC RAS and IZMIRAN that led to the Saint Petersburg magnetic observatory installation and development was set up by the joint efforts of the Laboratory of Geoinformatics and Geomagnetic Studies of the GC RAS, headed by Dr. A. Soloviev, and the Department of Geomagnetic Researches of the IZMIRAN SPb branch, directed by Prof. Yu. Kopytenko.

The magnetic survey of the observatory territory was performed by R. Sidorov and A. Grudnev. Magnetic hardware was

25 installed and set up for data registering by nearly all the authors of this pater except D. Kudin who later thoroughly controlled the magnetometer setup and monitored possible hardware failures.

Observatory infrastructure renovation, pavilion repair, power supply provision and azimuth mark construction was possible due to the orders of Yu. Kopytenko and A. Soloviev. Observatory hardware and software maintenance and renovation was performed by A. Kotikov and P. Sergushin and guided and assisted by A. Grudnev. All geodetic measurements for pillar

coordinate and reference azimuth determination were done by Dr. R. Krasnoperov.

All the absolute measurements at the Saint Petersburg observatory, that later were used to calculate the definitive magnetic data, and their operational calculation using the web services of the Russian-Ukrainian Geomagnetic Data Center, were



performed by A. Kotikov and P. Sergushin. A. Kotikov has been a regular staff member (magnetologist) of the observatory since its installation in 2012. Theodolite handover for its periodic servicing (verification and repair) was assisted by R. Sidorov and R. Krasnoperov.

The variation data preparation, its despiking, as well as baseline and definitive data calculation and their quality control was done by D. Kudin. The comparisons between the observatory data and the variation data from other observatories and the model data were performed by A. Soloviev and R. Sidorov.

# 6 Competing interests

The authors declare that they have no conflict of interest.

# 7 Acknowledgements

The results presented in this paper rely on data collected at the INTERMAGNET magnetic observatories. We express our gratitude to the national institutes that support them, we are grateful to the INTERMAGNET community for promoting the high standards of magnetic observatory practice (http://www.intermagnet.org) and the Russian-Ukrainian Geomagnetic Data Center (http://geomag.gcras.ru; re3data.org) for making the data freely available online. The research has been conducted in the framework of the Russian Science Foundation project No. 17-17-01215.

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



| IAGA code | Observatory name, country of location | Geogr. Lat | Geogr. Lon | Mag. Lat for 2015 | Mag. Lon for 2015 | Mag. Lat for 2016 | Mag. Lon for 2016 |
|---|---|---|---|---|---|---|---|
| BOX | Borok, Russia | 58.068° N | 38.233° E | 53.43°N | 123.53° E | 53.45°N | 123.56°E |
| LER | Lerwick, United Kingdom | 60.13° N | 1.18° W | 61.67°N | 88.63°E | 61.65°N | 88.64°E |
| NUR | Nurmijarvi, Finland | 60.51° N | 24.66° E | 57.79°N | 112.99°E | 57.80°N | 113.00°E |
| **SPG** | **Saint Petersburg, Russia** | **60.542° N** | **29.716° E** | **57.06°N** | **117.42°E** | **57.07°N** | **117.44°E** |

**Table 1 Reference information on the observatories whose variation records were chosen to be compared with the Saint Petersburg observatory data**





| IAGA code | Correlation coefficient between the component records | | | | | | | | | | | |
|---|---|---|---|---|---|---|---|---|---|---|---|---|
| | 17-18 March 2015 (storm period) | | | | 22-24 June 2015 (storm period) | | | | 1-4 June 2016 (quiet period) | | | |
| | X | Y | Z | F | X | Y | Z | F | X | Y | Z | F |
| BOX | 0.91 | 0.78 | 0.67 | 0.82 | 0.92 | 0.83 | 0.77 | 0.85 | 0.94 | 0.98 | 0.88 | 0.93 |
| LER | 0.86 | 0.61 | 0.51 | 0.72 | 0.83 | 0.49 | 0.47 | 0.69 | 0.72 | 0.71 | 0.65 | 0.78 |
| NUR | 0.99 | 0.89 | 0.81 | - | 0.99 | 0.89 | 0.95 | - | 0.99 | 0.99 | 0.96 | - |

**Table 2 Correlation coefficients for the magnetic field components from SPG and other observatories during the periods of disturbed (17-18 March and 22-24 June 2015) and quiet magnetic field (1-4 June 2016).**





| Model | Epoch | X | Y | Z |
|---|---|---|---|---|
| CHAOS-6 (CF) | 2015.1 | 14353.15 | 2615.35 | 50420.77 |
| CHAOS-6 (CF, AF85) | 2015.1 | 14424.07 | 2672.95 | 50293.85 |
| SIFM (CF) | 2015.1 | 14355.23 | 2616.28 | 50419.06 |
| EMM2015 (CF, AF720) | 2015.0 | 14513.75 | 2648.01 | 50346.38 |
| **SPG Definitive values** | **2015.1** | **14542.02** | **2551.72** | **50260.61** |
| **SPG Definitive values** | **2015.0** | **14543.68** | **2546.5** | **50262.27** |

**Table 3 Results of comparison between the average absolute values of the magnetic field vector components and the model values for different internal field models. CF stands for 'core field', and 'AF' for the anomalous (lithospheric) field; the number standing after the 'AF' in the model name indicates its spherical expansion order and degree.**

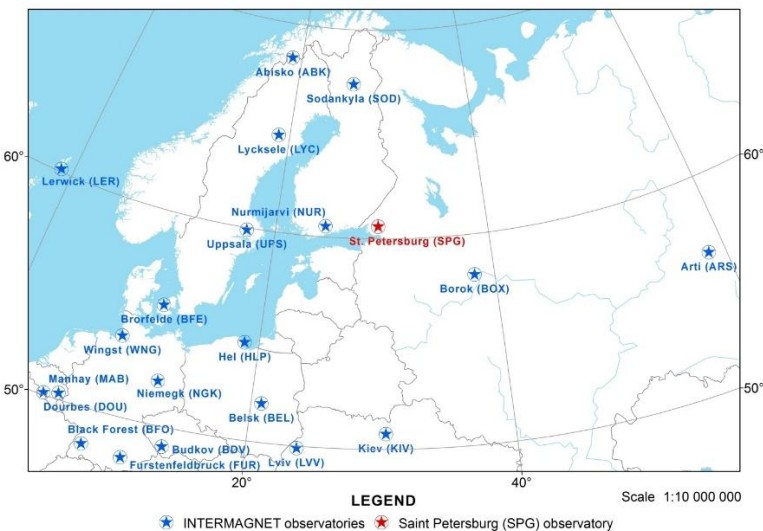

**Figure 1: Observatory geographical location. The Saint Petersburg observatory is marked with a red star. INTERMAGNET observatories are marked with blue stars.**





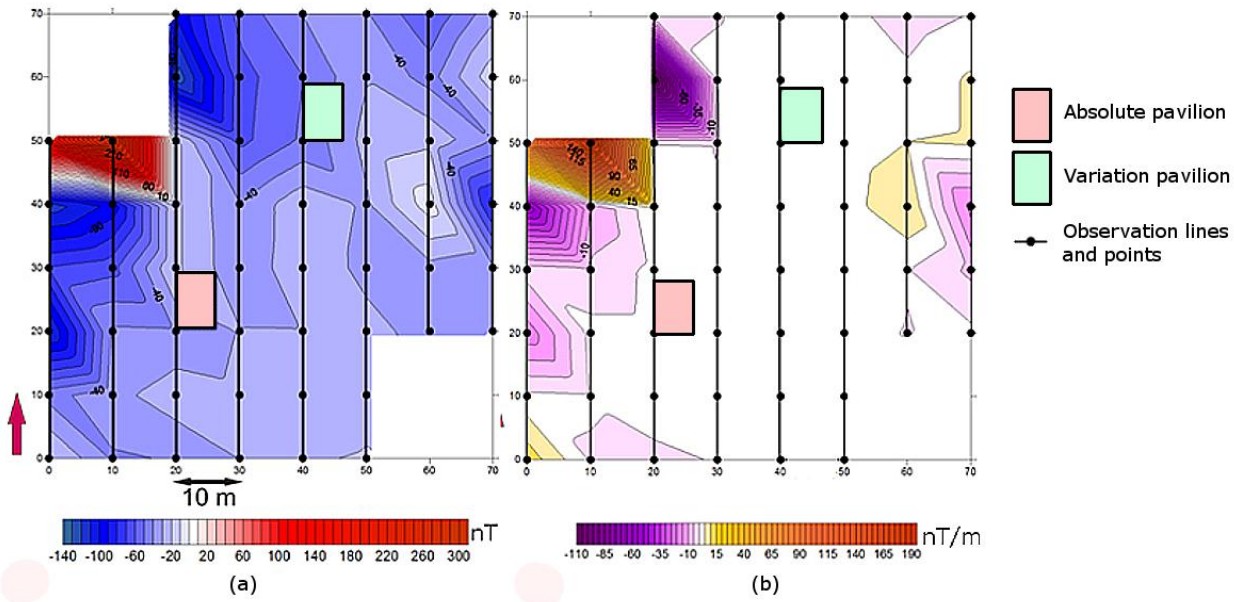

**Figure 2: Maps of magnetic anomalies (a) and vertical magnetic gradient (b) on the observatory territory.**

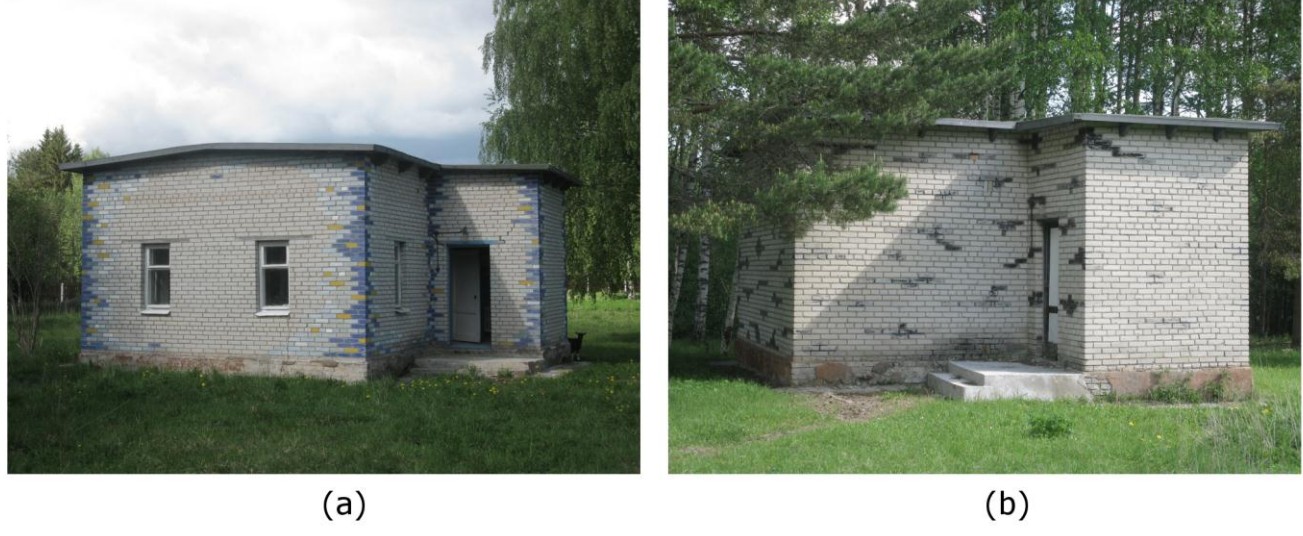

**Figure 3: Absolute (a) and variation (b) pavilions of the Saint Petersburg observatory.**





**Figure 4: Reference azimuth mark. The light bulb compartment (a), and the view of the mark from the absolute pavilion (b).**



**Figure 5: Minute magnetic variations of the X component during the strong magnetic storm of 17–18 March 2015.**

**Figure 6: Minute magnetic variations of the X component during the magnetic storm of 22–24 June 2015.**





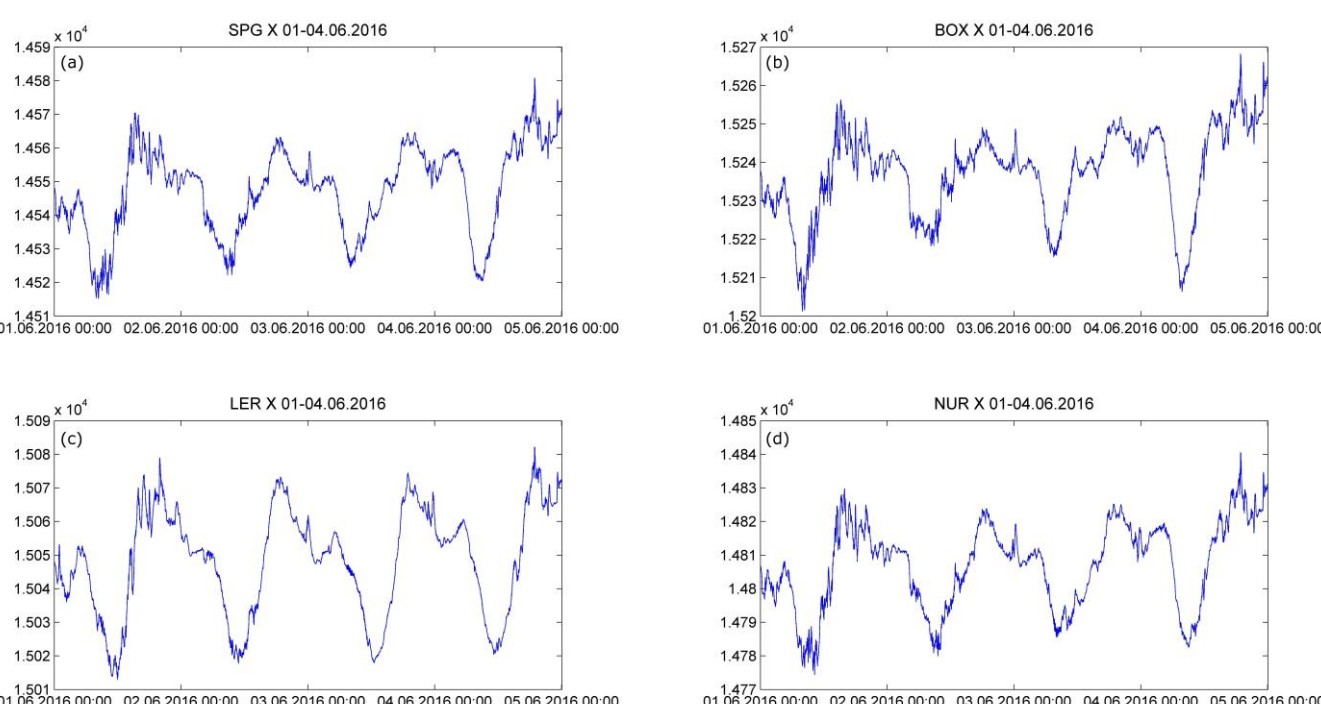

**Figure 7: Minute magnetic variations of the X component during a quiet period 01–04 June 2016**



Figure 8: Observed and adopted baseline values (a, b, c) for 2015 and absolute declination (d), inclination (e) and total intensity (f)





**Figure 9: Differences in the absolute measurements of the magnetic field components and definitive 1-minute values taken for the corresponding time moments: X component (a), Y component (b), Z component (c); fragments of the ΔF record over 2015 (d).**