# Peer review of "Saint Petersburg magnetic observatory: from Voeikovo subdivision to INTERMAGNET certification"

_Geoscientific Instrumentation, Methods and Data Systems, 2017_

## Referee Comment (RC1) · Anonymous Referee #1 · 5 Jul 2017

This is a useful reference for any data users on the history and improvement of the Saint Petersburg geomagnetic observatory with examples of data that provide information about the data quality and how the variations at that location compare to surrounding stations. The manuscript is generally well written and organised and in my opinion suitable for publication in GI. In some parts some more details would be useful. Below are my suggestions for improvement to the authors. (It seems that a few modifications have already been made to the version I was originally sent for review/assessment, so please just point this out in a reply if any of the below issues have already been addressed.)

[Figure]

a) I find the historical introduction how the present day data are linked to a long historical time series very useful, but I have difficulties to fully understand how much relocation has taken place and over which distances. How far away is Voiekovo from Pavlovs and Krasnoe Ozero? It might be useful to add a second panel to Fig. 1 with a map only of the surroundings of Saint Petersburg and the locations of these subsequent stations. The geographic coordinates of the observatory also would be useful information.

b) Text around Eq. 1: What was the sampling rate of the base station?

c) page 3, l. 20: clearly it would have been best to repeat the survey after removal of the source of disturbance. Although it sounds like a likely explanation that the gardening equipment caused the anomaly and the anomaly probably mostly vanished after its removal, has this in fact been checked?

d) Beginning of section 2.2: It would be informative to mention the material that was used for the pillars.

e) In section 2.3 the pillar difference should be discussed.

d) p.5, third paragraph: it could be useful to say more explicitly that the method that has to be used here is differential GPS

e) p.8,lines 7-8: "range of adopted baseline variability" sounds like the adopted baseline varies by those numbers over the year, which would be too much and clearly is not the case. As far as I understand the figure caption what is shown are the differences between baseline-corrected fluxgate recordings and the absolute values. Moreover, the larger values clearly look more like outliers (most likely in the absolute measurements), and rather than only giving the range it would make sense to also give the median or at least mention this important fact, that is clearly seen in the figure, in the text.

f) What are the short blue lines in Fig. 8?
g) p. 8, l. 23: I'm not convinced that a good agreement between data and models has really been demonstrated by these few numbers. At least the statement is too general, it should be discussed why differences >100nT can be understood as good agreement in this case.

Technical details: -p.5, l. 29 and p. 6, l.15: correct the brackets around references (bracket only around year if it is included in the running text like this) -p.7, l. 21 replace "registering" with "recording" -p.7,l. 32 better say "baseline jumps" instead of "baseline splits" -p.8,l. 6 in my opinion the sentence would be easier to understand if "values" was replaced by "fluxgate recordings"

---

## Author Comment (AC1) · 1 Aug 2017

Dear Anonymous Referee, Thank you for your comment and for valuable remarks on the manuscript! The initial version contained some more details on the observatory equipment, magnetic survey results etc. and corresponding photos, but during the manuscript preparation we decided to make a short (but informative) paper with the general aspects, and excluded some details. We shall take your remarks into account.

a) According to your advice we've prepared a map of Saint Petersburg city vicinity including Voeikovo and Pavlovsk, and this map will be the second section of the first figure (please have the figure enclosed). We have also included the current coordinates

of the observatory in the manuscript.

b) The sampling rate of the base station was 3 seconds (we used the GSM-19 proton magnetometer installed at the absolute pavilion as a base station).

c) Unfortunately, the survey was not repeated (but it will surely be repeated in future).

d) The material is marble, and the tops were fixed on the marble pillars using grout.

e) Certainly the influence of some elements of the pavilion interior (like the power cables etc) and some influence of natural disturbances could cause such a difference of 5.5 nT, but the point is that this value is quite constant. After we repeated the determination of the pillar difference in 2014 or 2015, this value was generally the same except for some second decimal places. So it indicates the stability of the field distribution on the absolute pavilion area. Generally, this difference is less or more depending on the particular observatory (in a particular location) and the particular pillar, and on some observatories the difference between the observation pillar and the scalar magnetometer pillar can be even up to 30 nT but it does not significantly change from measurement to measurement.

d) We'll provide the information on the point positioning and the GPS differential mode.The GPS receivers were positioned at auxiliary points for determining the azimuth of the baseline between these points. Point positioning was performed in differential mode. Two continuously operating GPS-stations SVTL and PULK, located within 150 km from the observatory site, were used as base stations.

e) Yes indeed, probably we should formulate it this way: "The corresponding differences were calculated; their dispersion plots are given in Fig. 9a, b, c. As seen, some outlier values produced the large differences varying from –7.98 to 18.24 nT, from –7.09 to 15.86 nT, and from –0.78 to 4.23 nT for X, Y, and Z component, respectively. After the removal of the outliers, the obtained RMS deviations for baselines for the period 01.01.2015–01.01.2016 were 2.91, 2.08 and 0.61 nT for X, Y, and Z, respectively."
f) The short blue lines on the right of the plots indicate the adopted baselines calculated for the next year (they include the baselines from December 1 of the current year); this image was generated by our data visualization system. I added the caption as an element of the legend.

g) We suppose that the differences of even more than 100 nT still allow to make a conclusion that the magnetometer set is properly installed and the component and total field data is physically close, as the values are close in general (thousands of nT). The differences in hundreds of nT can be caused by particular anomalies of geological origin in the vicinity of the observatory location whose effect can produce a constant component, and the contribution of these anomalies could be not properly reflected in the model data. Moreover, the models selected for the comparison, representing the internal magnetic field, do not generally reflect exactly the same distributions of the magnetic anomalies due to possible differences in their compilation and the source data used, and one can clearly see the differences between the model values up to even 150 nT for the same epoch.

Thank you again very much for your comment and for the mentioning of technical details to be improved.
* * *
(a)

Abisko (ABK)

Sodankyla (SOD)

Lycksele (LYC)

60°                                                              60°

Lerwick (LER)

Nurmijarvi (NUR)

Uppsala (UPS)    St. Petersburg (SPG)

Arti (ARS)

Borok (BOX)

Brorfelde (BFE)

Wingst (WNG)    Hel (HLP)

Manhay (MAB)

50°    Dourbes (DOU)    Niemegk (NGK)                          50°

Black Forest (BFO)    Belsk (BEL)

Budkov (BDV)    Kiev (KIV)
Furstenfeldbruck (FUR)    Lviv (LVV)

20°                                          40°

**LEGEND**                                   Scale  1:10 000 000

⭐ INTERMAGNET observatories  ⭐ Saint Petersburg (SPG) observatory

(b)

St. Petersburg (SPG)

*Lake Ladoga*

*Gulf of Finland*

60°                                                              60°

Voeikovo

Saint Petersburg

Pavlovsk

30°

**LEGEND**                                   Scale 1:500 000

⭐ Historical observatories  ⭐ Saint Petersburg (SPG) observatory    10 5 0    10    20 Km

**Fig. 1.**

[Figure]

**Fig. 2.**

---

## Referee Comment (RC2) · L. Hegymegi (Referee) · 14 Aug 2017

This paper gives a good example how to renew an observatory starting from the beginning and shows step by step the whole procedure. This work is nearly the same as the foundation of a new observatory. It shows which are the necessary actions to be done inside and around the location of instrument huts and on all the territory of the observatory. It gives a good example how to build an azimuth reference mark for absolute observations and and how to determine its direction. The paper can be a useful material for those people who do the same job or builds and installs a new observatory. The paper is good for publication in GI.

[Figure]

I have some comments and questions to the authors.

- In 2.2 mentioned former heating system used copper wires. Was it dismounted on only renewed? If it is still in use a special attention has to be given to its structure because temperature differences at the connections of the copper tubes can generate thermal currents and consequently slowly changing magnetic field. This field gives an addition to the geomagnetic field. If warm water is used for heating it is better solution to apply plastic tubes.

- In 2.3 the power system with underground cables is mentioned. It is an important issue to use short cables if possible and it is useful to apply overvoltage protection against induction peaks caused by lightnings. - The same is valid for data lines.

- Why you did not used UPS instead of LER for data analysis? It is much closer to SPG in longitude.

- For raw instrument check the data analysis as described is good but more precise data can only be obtained only if another recording instrument is installed at the same place. Will it be possible in the future?

---

## Author Comment (AC2) · 25 Aug 2017

Dear László, Thank you very much for your referee comment on our manuscript. We'll consider the issues you mentioned and update our paper with the details on them. Now we are glad to answer to your comments and questions.

1. On the copper wires of the heating system. Currently the copper heating system is not used. We'll mention this in the section 2.2.

2. On the underground cable lines. The cables used for power supply for the magnetometers, as well as the data cables, are not so long, and of course they are protected

by the lightning protection modules. We'll also mention this in the text.

3. On the data comparison between SPG and other observatories. Initially we performed the correlation calculation for the data from even more magnetic observatories along the magnetic latitude of the SPG. I supposed that the display of the correlation between the SPG and NUR data is enough (as UPS is close to NUR too), and the comparison between SPG and LER which is farther from this latitude was a sort of experiment. Now, after your comment, we have calculated the correlations between SPG and UPS and we'll provide the corresponding rows in the tables for correlation coefficients.

4. On the data check using another magnetometer at the same location. Yes, in future such analysis will be possible. In June 2017 we installed a new POS-4 vector Overhauser magnetometer at the one of supplementary pavilions at the SPG observatory in a test mode (later it will be relocated to another station), and the provisional comparison between the H, Z and F data from POS-1 and from the FGE and GSM magnetometers showed generally good agreement; later we'll analyze and compare the data for a longer period.

Thank you again for your opinion on the manuscript and your useful comments and advice.
* * *